# Relaxed Octahedral Group Convolution for Learning Symmetry Breaking in 3D Physical Systems

**Rui Wang**
Massachusetts Institute of Technology
rayruw@mit.edu

**Robin Walters**
Northeastern University
r.walters@northeastern.edu

**Tess E.Smidt**
Massachusetts Institute of Technology
tsmidt@mit.edu

## Abstract

Deep equivariant models use symmetries to improve sample efficiency and generalization. However, the assumption of perfect symmetry in many of these models can sometimes be restrictive, especially when the data does not perfectly align with such symmetries. Thus, we introduce relaxed octahedral group convolution for modeling 3D physical systems in this paper. This flexible convolution technique provably allows the model to both maintain the highest level of equivariance that is consistent with data and discover the subtle symmetry-breaking factors in the physical systems. Empirical results validate that our approach can not only provide insights into the symmetry-breaking factors in phase transitions but also achieves superior performance in fluid super-resolution tasks.

## 1 Introduction

Symmetry and equivariance play pivotal roles in the advancement of deep learning [60, 28, 4, 8, 52, 59, 7]. Specifically, equivariant convolution [7, 12] and graph neural networks [40, 45, 6], which integrate symmetries into the design of the architectures, have demonstrated notable success in modeling complex data. By constructing a model inherently equivariant to the transformations of relevant symmetry groups, we ensure automatic generalization across these transformations. This enhances not only the model's robustness to distributional shifts but also its sample efficiency [19, 48, 31]. Furthermore, Noether's theorem establishes a relationship between conserved quantities and symmetry groups [29]. Consequently, neural networks that preserve these symmetries are poised to generate physically more accurate predictions [51, 50, 22].

A limitation of many existing equivariant models is they assume that data has perfect symmetry matching the models' equivariance. This means they are learning functions that are strictly invariant or equivariant under given group actions. However, this can be overly restrictive, especially when the data does not possess perfect symmetry or exhibits less symmetry than what the model accounts for. For instance, [49] indicates that imposing incorrect or irrelevant symmetries may hurt performance. [44] found that the equivariant models fail to learn when the output has lower symmetry than the input and the models themselves. [51] empirically shows that approximately equivariant models outperform models with no symmetry bias and those with strict symmetry in learning fluid dynamics.

Consequently, several recent works have focused on relaxing strict equivariance constraints and designing approximately equivariant networks. For instance, [10] showed that relaxing strict spatial weight sharing in conventional 2D convolution can improve image classification accuracy. [14] suggested substituting equivariant layers with a combination of those layers and non-equivariant

MLP layers for modeling soft equivariances. Lastly, [51] proposed a relaxed group convolution that introduces additional relaxed weights into regular group convolution. The relaxed weights are additional trainable parameters that are initialized as equal but can vary across group elements during training to break to strict equivariance of the models.

In this paper, we unveil the ability of relaxed group convolution for modeling 3D physical systems. We highlight its ability to consistently capture the correct symmetry inductive biases from the data and illustrate that the relaxed weights post-training not only are interpretable but also can pinpoint symmetry-breaking factors. More specifically, our contributions include:

- We develop a novel layer, the relaxed octahedral group convolution, which is approximately equivariant to $\mathbb{R}^3 \rtimes O$, for modeling 3D physical systems with symmetry breaking.
- We show theoretically that the relaxed weights learn to break symmetry only when necessitated by training data. By examining the relaxed weights post-training, we determine the symmetry-breaking factors in the intermediate structures of $BaTiO_3$ at varying temperatures.
- We demonstrate our method outperforms baselines with no symmetry bias and with overly strict symmetry on the task of fluid super-resolution for both channel flow and isotropic flow.

## 2 Background

### 2.1 Symmetry and Equivariant Neural Networks

**Equivariance and Invariance**    We say a function $f \colon X \to Y$ is $G$-**equivariant** if

$$f(\rho_{\text{in}}(g)(x)) = \rho_{\text{out}}(g)f(x) \tag{1}$$

for all $x \in X$ and $g \in G$. $\rho_{\text{in}}$ is the input group representation of $G$ acting on $X$ and $\rho_{\text{out}}$ is the output group representation that acts on $Y$. The function $f$ is $G$-**invariant** if $f(\rho_{\text{in}}(g)(x)) = f(x)$ for all $x \in X$ and $g \in G$. This is a special case of equivariance for the case $\rho_{\text{out}}(g) = 1$.

**Equivariant Neural Networks**    The key to building equivariant neural networks lies in the principle that the composition of equivariant functions remains equivariant. Thus, a neural network will be strictly equivariant as long as all of its layers, activation functions, pooling, and normalization are equivariant. The primary challenge in this field centers on developing trainable equivariant linear layers, often involving techniques like weight sharing and weight tying across group elements. Our focus in this paper is on group convolution [8] and its applications to three-dimensional data structured on regular meshes.

**Octahedral Group**    The octahedral group, often denoted as $O$, is a finite subgroup of $O(3)$ that describes all the symmetries of a regular octahedron or a cube. While not the largest finite subset of $O(3)$ (the largest one is the icosahedral group), the octahedral group, with its 24 rotations or 48 total elements including reflections, offers a substantial representation of 3D symmetries and is still relatively efficient to compute. 3D data is often structured on a cubic grid, making the octahedral group potentially more suitable for real-world 3D data compared to other finite subgroups of $O(3)$. More importantly, octahedral symmetries are compatible with the cubic lattice and relevant to many physical systems, such as certain crystal structures [33, 47, 55] and fluid systems [21, 50].

### 2.2 Regular Group Convolution

Group convolution achieves equivariance by sharing the kernel weights across all group elements. The convolutional layers of traditional CNNs are nothing else but group convolution specialized to the translation group $\mathbb{Z}^2$ or $\mathbb{Z}^3$. By replacing the spatial shifts with transformations from a more general group $G$, we can make models become equivariant to other groups.

**Lifting Convolution**    To build $G$-group convolution networks, the initial step typically involves lifting input to a function on $G$. It maps input signals $f_{\text{in}} \colon \mathbb{Z}^3 \to \mathbb{R}$ to output signals $f_{\text{out}} \colon G \to \mathbb{R}$ on arbitrary group $G = (\mathbb{Z}^3, +) \rtimes H$. A group action from $G$ can be split into a translation $L_{\mathbf{x}}$ and a transformation $L_h$ from the subgroup $H$.

$$f_{\text{out}}(\mathbf{x}, h) = (f_{\text{in}} * \psi)(\mathbf{x}) = \sum_{\mathbf{y} \in \mathbb{Z}^3} f_{\text{in}}(\mathbf{y}) L_h L_{\mathbf{x}} \psi(\mathbf{y}) = \sum_{\mathbf{y} \in \mathbb{Z}^3} f_{\text{in}}(\mathbf{y}) L_h \psi(\mathbf{y} - \mathbf{x}) \tag{2}$$

So the lifting convolution is the same as conventional convolutions with a stack of filter banks transformed according to the subgroup $H$.

**Group Convolution**  After the lifting convolution layer, both the filter and input are now functions on $G = (\mathbb{Z}^3, +) \rtimes H$. A $G$-equivariant group convolution then takes as input a $c_{\text{in}}$-dimensional feature map $f_{\text{in}} : G \to \mathbb{R}^{c_{\text{in}}}$ and convolves it with kernel $\Psi : G \to \mathbb{R}^{c_{\text{out}} \times c_{\text{in}}}$ over a group $G$,

$$f_{\text{out}}(\mathbf{x}, h) = (f_{\text{in}} * \Psi)(\mathbf{x}, h) = \sum_{\mathbf{y} \in \mathbb{Z}^3} \sum_{h' \in H} f_{\text{in}}(\mathbf{y}, h') \Psi(h^{-1}(\mathbf{y} - \mathbf{x}), h^{-1}h') \tag{3}$$

The last layer usually averages over the $h$-axis and outputs a function on $\mathbb{Z}^3$.

# 3  Methodology

## 3.1  Relaxed Octahedral Group Convolution

Based on the Eqn.5 in [51], we first define Relaxed Octahedral Group Convolution $f \tilde{\star}_{\mathbb{Z}^3 \rtimes O} \Psi$ that is both $\mathbb{Z}^3$-translation equivariant and (approximately) $O$-equivariant as below.

$$\begin{aligned}
f_{\text{out}}(\mathbf{x}, h) = (f_{\text{in}} \tilde{\star} \Psi)(\mathbf{x}, h) &= \sum_{\mathbf{y} \in \mathbb{Z}^3} \sum_{h' \in O} f_{in}(\mathbf{y}, h') \Psi(\mathbf{y} - \mathbf{x}, h, h') \\
&= \sum_{\mathbf{y} \in \mathbb{Z}^3} \sum_{h' \in O} f_{\text{in}}(\mathbf{y}, h') \sum_{l=1}^{L} w_l(h) \Psi_l(h^{-1}(\mathbf{y} - \mathbf{x}), h^{-1}h')
\end{aligned} \tag{4}$$

Unlike a strictly equivariant group convolution layer, the above operation uses several filters $\{\Psi_l\}_{i=l}^{L}$ instead of one shared filter. Since the scalar coefficients $w_l(h)$ can depend on the specific group element $h$ and not just the difference $h^{-1}h'$ this breaks equivariance unless the $w_l(h)$ are constant for all $h \in O$. $w_l(h)$ is not necessarily same for all $l$ to hold equivariance. In fact, we initialize the $w_l(h)$ this way so the model will be perfectly equivariant at initialization. When data exhibits broken symmetry, the $w_l(h)$ learn to be unequal to break the strict equivariance and adapt to the data. We will prove this in the next subsection.

## 3.2  Efficient Implementation with Separable Convolutions

Though group convolution has been a powerful tool for many applications, its computational complexity scales exponentially with the dimensionality of the group, which is the main practical challenge when dealing with the octahedral group. To improve parameter efficiency, we employ separable group convolution that assumes the convolution kernels are separable [27]. That is $\Psi_l(\mathbf{x}, h) = \Psi_l^T(\mathbf{x}) \Psi_l^O(h)$ with $\Psi_l^T$ stays constant along $O$ and $\Psi_l^O$ stays constant along $\mathbb{Z}^3$. Then Eqn. (4) can be rewritten as:

$$\begin{aligned}
(f_{in} \tilde{\star} \Psi)(\mathbf{x}, h) &= \sum_{\mathbf{y} \in \mathbb{Z}^3} \sum_{h' \in O} f_{\text{in}}(\mathbf{y}, h') \sum_{l=1}^{L} w_l(h) \Psi_l^O(h^{-1}h')) \Psi_l^T(h^{-1}(\mathbf{y} - \mathbf{x})) \\
&= \sum_{l=1}^{L} w_l(h) \left[ \sum_{\mathbf{y} \in \mathbb{Z}^3} \left[ \sum_{h' \in O} f_{\text{in}}(\mathbf{y}, h') \Psi_l^O(h^{-1}h')) \right] \Psi_l^T(h^{-1}(\mathbf{y} - \mathbf{x})) \right]
\end{aligned} \tag{5}$$

By using separable convolution, the number of parameters in a single kernel goes from $|O| \times S^3$ to $|O| + S^3$, where $S$ is the kernel size. Intuitively speaking, $\Psi_l^T(\mathbf{x}) \Psi_l^O(h)$ is rank-1 tensor decomposition of $\Psi_l(\mathbf{x}, h)$. Note that we only relax the subgroup $O$ in this paper rather than the entire group $G$, as relaxing the 3D translation group would be too computationally intensive.

## 3.3  Finding Symmetry Breaking Factors with Relaxed Group Convolution

In the relaxed group convolution, the initial relaxed weights $w_l(h)$ are set to be the same, ensuring that the model exhibits equivariance prior to being trained. In this subsection, we prove that these relaxed weights only deviate from being equal only when the symmetry of the output is less than that of the model.

**Proposition 3.1.** *Consider a relaxed group convolution neural network where the relaxed weights in each layer are initialized to be identical to maintain $G$-equivariance. If it is trained to map an input $X$ to the output $Y$, its relaxed weights will learn to be distinct across group elements in $G$ during training in a way such that it is equivariant to $G \cap Stab(X) \cap Stab(Y)$, which is the intersection of the stabilizers of the input and the output and $G$.*

Proof can be found in the Appendix A.

### 3.4 Finding Symmetry Breaking Factors in a Simple 2D Example

This useful property allows us to discover symmetry or identify the symmetry-breaking factors in the data because the relaxed weights can tell us whether a transformation $h$ stabilizes the output $Y$. We provide a clear illustration of this through a simple 2D example.

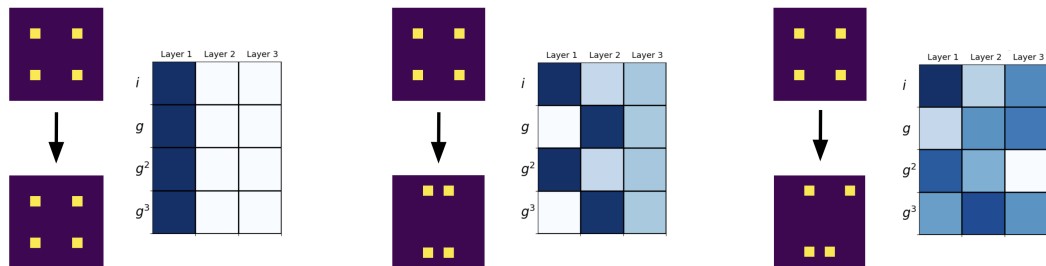

Figure 1: Visualization of tasks and corresponding relaxed weights after training. A 3-layer $C_4$-relaxed group convolution network with $L = 1$ is trained to perform the following three tasks: 1) map a square to a square; 2) deform a square into a rectangle; 3) map a square to a non-symmetric object.

We trained a 3-layer $C_4$ relaxed group convolution network with $L = 1$ on the following three tasks: 1) map a square to a square; 2) deform a square into a rectangle; 3) map a square to a non-symmetric object. Both the input and output are single-channel images. As shown in Figure 1, in the first task where the output is a square with $C_4$ symmetry, relaxed weights across all layers remain equal throughout training. For the second task, where the output is a rectangle exhibiting $C_2$ symmetry, the relaxed weights learn to be different. However, the weights corresponding to the group elements $i$ and $g^2$ are the same, as do the weights for $g$ and $g^3$. That implies that the output is invariant to 180-degree rotations and the model becomes equivariant to $C_2$. In the final task, where the output lacks any meaningful symmetries, the relaxed weights diverge entirely for the four group elements, thereby breaking the model's equivariance.

When dealing with larger groups, decomposing relaxed weights—which can be interpreted as signals on the group—into irreps can simplify the task of pinpointing broken symmetries. By projecting the relaxed weights onto the irreducible representations (irreps) of the group, similar to calculating its Fourier components, one can assess which symmetries are preserved or broken. More specifically, if the signal were perfectly symmetric under the group operations, you would expect non-zero contributions only from the trivial representation and other irreps that align with the signal's symmetries. Any significant contributions from other irreps suggest that those particular symmetries are broken. Take, for example, the relaxed weights from a neural network layer illustrated in Figure 1, projected onto $C_4$'s four one-dimensional irreps. For the first task, only the Fourier component for the trivial representation is non-zero. In the second task, Fourier components for both the trivial and the sign representation are non-zero, suggesting that either 90 or 180-degree rotational symmetries are broken. Since the remaining irreps are zero, we can conclude the output is still invariant under 180-degree rotations. In the third task, all Fourier components are non-zero. This is related to the approach that uses trainable irreps to discover symmetry-breaking parameters in this paper [44].

In a word, the relaxed group convolution has the potential to discover the symmetry in the data, while also reliably preserving the highest level of equivariance that is consistent with data.

# 4 Related Work

## 4.1 Equivariance and Invariance

Symmetry has been subtly integrated into deep learning to build networks with specific invariances or equivariances. Convolutional neural networks revolutionized computer vision by using translation equivariance[60, 61], while graph neural networks exploit permutation symmetries [41, 26]. Equivariant deep learning models have excelled in image analysis[8, 5, 6, 52, 28, 2, 56, 7, 13, 53, 9, 20], and their application is now expanding to physical systems due to the deep relationship between physical symmetries and the principles of physics. For instance, [50] designed fully equivariant convolutional models with respect to symmetries of scaling, rotation, and uniform motion, particularly in fluid dynamics scenarios. [23] introduced Steerable Conditional Neural Processes to learn the complex stochastic processes in physics while ensuring that these models respect both invariances and equivariances. [22] equivariant Fourier neural operator to solve partial differential equations by leveraging the equivariance property of the Fourier transformation. Additionally, the domain of graph neural networks has seen a surge in the development of equivariant architectures, especially for tasks related to atomic systems and molecular dynamics. This growth is attributed to the inherent presence of symmetries in molecular physics, such as roto-translation equivariance in the conformation and coordinates of molecules. [1, 39, 45, 32, 18, 62, 42, 36]. For instance, [30, 31] proposed Equiformers for modeling 3D atomistic graphs. They are graph neural networks leveraging the strength of Transformer architectures and incorporating equivariant features with E3NN [19].

## 4.2 Approximate Symmetry and Symmetry Breaking

Many real-world data rarely conform to strict mathematical symmetries, due to noise and missing values or symmetry-breaking features in the underlying physical system. Thus, there have been a few works trying to relax the strict symmetry constraints imposed on the equivariant networks. [10] first showed that relaxing strict spatial weight sharing in conventional 2D convolution can improve image classification accuracy. [51] generalized this idea to arbitrary groups and proposed relaxed group convolution, which is biased towards preserving symmetry but is not strictly constrained to do so. The key idea is relaxing the weight-sharing schemes by introducing additional trainable weights that can vary across group elements to break the strict equivariance constraints. [37] further provides a theoretical study of how the data equivariance error and the model equivariance error affect the models' generalization abilities. In this paper, we further extend relaxed group convolution to three-dimensional cases and reveal its potential in symmetry discovery problems. Additionally, [14] proposed a mechanism that sums equivariant and non-equivariant MLP layers for modeling soft equivariances, but it cannot handle large data like images or high-dimensional physical dynamics due to the number of weights in the fully-connected layers. [24] formalizes active and approximate symmetries in graph neural nets that operate on a fixed graph domain, highlighting a tradeoff between expressiveness and regularity when incorporating these symmetries.

## 4.3 Super-resolution for fluid flows

Refining low-resolution images using super-resolution techniques is essential for various applications in computer vision [35, 46, 34, 38]. Due to the enormous computational cost of generating high-fidelity simulations, a variety of deep learning models based on MLPs [11], CNNs[15, 17, 43], and GANs [54, 57] have been proposed for the super-resolution fluid dynamics [16]. The closest work to ours is [58] which applies 2D rotationally equivariant CNNs to upscale the velocity fields of fluid dynamics. However, they do not consider the scenarios of approximate symmetry and three-dimensional fluid dynamics as we do.

# 5 Experiments

## 5.1 Discover Symmetry Breaking Factors in Phase Transitions

**Phase Transition**  Modeling a phase transition from a high symmetry (like octahedral) to a lower symmetry is a common topic of interest in the fields of materials science. This change can be understood as a transformation in the arrangement or orientation of atoms within a crystal lattice. For instance, perovskite structures are a class of materials with the general formula ABO3. The A and B

are cations of different sizes, and O is the oxygen anion. The B cation is typically a transition metal. Under certain conditions like a decrease in temperature, these BO6 octahedra may undergo distortion.

To illustrate, consider the case of Barium titanate ($BaTiO_3$), as shown in Figure 2. At high temperatures, $BaTiO_3$ has a cubic perovskite structure with the Ti ion at the center of the octahedron. As one progressively cools $BaTiO_3$, it undergoes a series of symmetry-breaking phase transitions. Initially, around 120°C, there is a shift from the cubic phase to a tetragonal phase, where the Ti ion is displaced from its central position. In a tetragonal system, two of the axes are of equal length and the unit cell is in the shape of a rectangular prism where the base is a square. This is followed by a transition to orthorhombic at about -90°C, where all three axes are of different lengths.

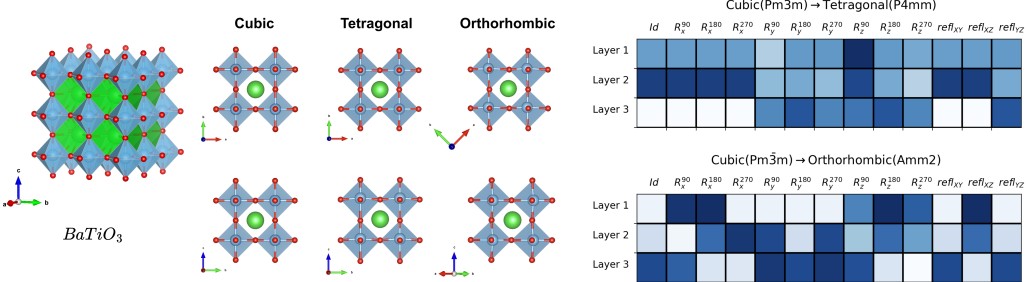

Figure 2: Left: Visualization of $BaTiO_3$: As temperature decreases, it undergoes a series of symmetry-breaking phase transitions, transitioning from a cubic structure to a tetragonal phase, and eventually to an orthorhombic form. Right: Visualization of relaxed weights of 16 of a total 48 elements from the two 3-layer models trained to predict the tetragonal and orthorhombic structures from a cubic system, including rotations along $x$,$y$,$z$ axes and reflections over $XY$, $YZ$, $XZ$ planes.

**Experimental Setup** We download the fractional coordinates of BaTiO3 in cubic, tetragonal, and orthorhombic phases from the Material Project[1]. We use 3D tensors to represent these systems where the pixels corresponding to atoms are non-zero. We train a 3-layer relaxed group convolution network to 1) map the cubic system to the tetragonal system; and 2) map the cubic system to the orthorhombic system until overfitting.

**Find Symmetry Breaking Factors Using Relaxed Weights** Figure 2 visualizes the relaxed weights from the two 3-layer models trained to predict the tetragonal and orthorhombic structures from a cubic system. We only show the relaxed weights corresponding to the rotations along the $x$, $y$, and $z$ axes and reflections over $XY$, $YZ$, and $XZ$ planes as they are more straightforward to understand. As shown in Figure 2, when the model is trained to predict the tetragonal system, the post-training relaxed weights successfully find that the four-fold rotation symmetries along $y$ and $z$ axes are broken. When the model is trained to predict the orthorhombic system, only two-fold rotation symmetry along the $y$ axis and reflection symmetries over $XY$ and $YZ$ planes are left, because the relaxed weight of $R_y^{90}$ and $R_y^{270}$ are the same and refl$_{XY}$ and refl$_{YZ}$ are the same as the identity. This aligns with the space group $Amm2$ of orthorhombic crystal system [3]. Such results highlights the capability of relaxed group convolution to automatically discover symmetries and symmetry-breaking factors.

### 5.2 Super-resolution of Velocity Fields in Three-dimensional Fluid Dynamics

**Data Description.** We use the direct numerical simulation data of the channel flow ($2048 \times 512 \times 1536$) turbulence and the forced isotropic turbulence ($1024^3$) from Johns Hopkins Turbulence Database [25]. For each dataset, we acquire 50 frames of velocity fields, which are then downscaled by half and segmented into $64^3$ cubes for experimental use. These cubes are further downsampled by a factor of 4 to serve as input for our superresolution model. The models are trained to generate $64^3$ simulations from $16^3$ downsampled versions of them. Because of the spatial weight sharing of CNNs, we can apply our model to 3D input with any resolutions during inference.

**Experimental Setup** Figure 3 visualizes the model architecture we use for super-resolution. We evaluate the performance of Regular, Group Equivariant, and Relaxed Group Equivariant layers built

---

[1]Material Project: `https://next-gen.materialsproject.org/`

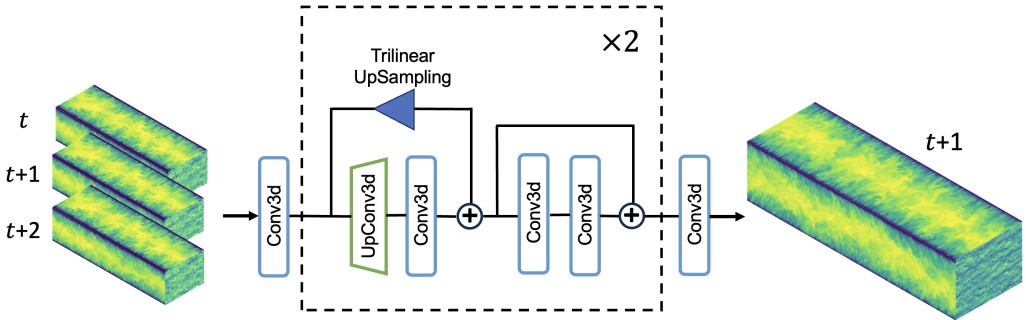

Figure 3: The architecture of the super-resolution model includes an input layer, an output layer, and eight residual blocks. Since it has two layers of Transoposed Convolution(UpConv3d), the model produces simulations that are upscaled by a factor of four.

into this architecture in the tasks of upscaling channel flow and isotropic turbulence. The models take three consecutive steps of downsampled $16^3$ velocity fields as input and predict a single step of $64^3$ simulation, enabling them to infer vital attributes like acceleration and external forces for precise small-scale turbulence predictions. We use the L1 loss function over the L2 loss, as it significantly enhances performance. We split the data 80%-10%-10% for training-validation-test across time and report mean absolute errors over three random runs. As for hyperparameter tuning, except for fixing the number of layers and kernel sizes, we perform a grid search for the learning rate, hidden dimensions, batch size, and the number of filter bases for all three types of models.

Table 1: Prediction MAE of trilinear upsampling, non-equivariant, equivariant, relaxed equivariant models on the super-resolution of channel flow and isotropic flow.

| Model | Channel Flow $(10^{-2})$ | | | | Isotropic Flow $(10^{-1})$ | | | |
|---|---|---|---|---|---|---|---|---|
| | Trilinear | Conv | Equiv | R-Equiv | Trilinear | Conv | Equiv | R-Equiv |
| MAE | 5.241 | 2.602 | 2.540 | **2.441** | 5.248 | 1.215 | 1.119 | **1.000** |

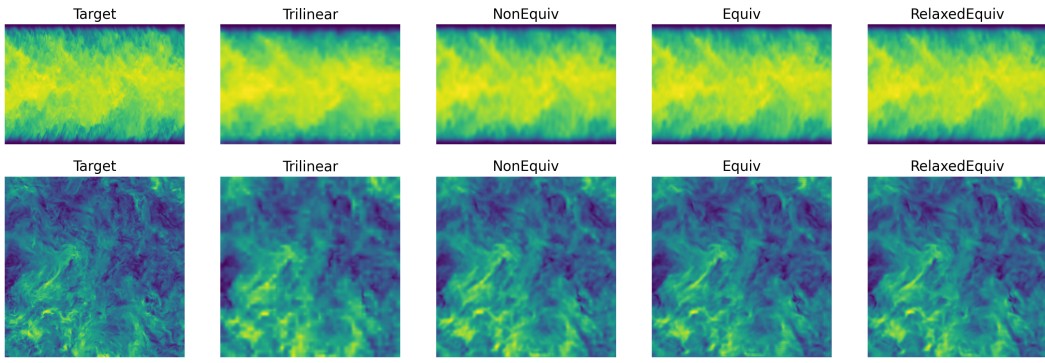

Figure 4: Prediction visualization of a cross-section along the z-axis of the velocity norm fields.

**Prediction Performance**  Table 1 shows the prediction MAE of trilinear upsampling, non-equivariant, equivariant, and relaxed equivariant models applied to super-resolution tasks for both channel and isotropic flows. Figure 4 shows the 2D velocity norm field of predictions. As we can see, imposing equivariance and relaxed equivariance consistently yield better prediction performance.

Figure 5 visualizes of relaxed weights of the first two layers from the models trained on isotropic flow and channel flow. relaxed weights for isotropic flow stay almost the same across group elements while those for channel flow vary a lot. This suggests that the relaxed group convolution can discover symmetry in the data even if the symmetry lies in the sample space, rather than individual samples.

For isotropic turbulence, even if individual samples might not seem symmetrical, the statistical properties of their velocity fields over time and space are invariant with respect to rotations. This makes models trained on isotropic flow benefit more from the equivariance, as shown in the table. The channel flow, on the other hand, is driven by a pressure difference between the two ends of the channel together with the walls, which makes the turbulence inherently anisotropic. In such cases, the relaxed group convolution is preferable, as it adeptly balances between upholding certain symmetry principles and adapting to factors that introduce asymmetry.

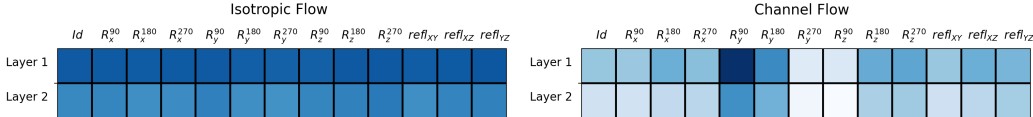

Figure 5: Visualization of the relaxed weights of first two layers from the models trained on the isotropic flow and channel flow.

# 6    Discussion

We propose 3D Relaxed Octahedral Group Convolution Networks that avoid stringent symmetry constraints to better fit real-world three-dimensional physical systems. We demonstrate its ability to consistently capture the highest level of equivariance that is consistent with data and highlight that the relaxed weights can discover the symmetry and symmetry-breaking factors. Future works include the theoretical study of the relaxed group convolution discovering symmetries in the sample space instead of individual samples. Additionally, we aim to uncover other potential applications of our model in phase transition analysis and material discovery.

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

# Appendix

## A   Theoretical Analysis

**Proposition A.1.** *Consider a relaxed group convolution neural network where the relaxed weights in each layer are initialized to be identical to maintain G-equivariance. If it is trained to map an input $X$ to the output $Y$, its relaxed weights will learn to be distinct across group elements in $G$ during training in a way such that it is equivariant to $G \cap Stab(X) \cap Stab(Y)$, which is the intersection of the stabilizers of the input and the output and $G$.*

*Proof.* Let $G$ be the semi-direct product of the translation group $(\mathbb{Z}^3, +)$ and a group $H$. Without loss of the generality, we only consider the composition of one lift convolution layer and a relaxed group convolution layer with a single filter bank case (i.e. $L = 1$). Suppose the input is $f_0(\boldsymbol{x})$ and $\phi$ is an unconstrained kernel, then the output of the lift convolution layer is:

$$f_1(\boldsymbol{y}, h) = \sum_{\boldsymbol{x} \in \mathbb{Z}^3} f_0(\boldsymbol{x})\phi(h^{-1}(\boldsymbol{x} - \boldsymbol{y})), \ \ h \in G$$

Now we prove that $f_1(\boldsymbol{y}, h) = f_1(k\boldsymbol{y}, kh), \ k \in G$ only when $k$ stabilizes $f_0$.

$$
\begin{aligned}
f_1(k\boldsymbol{y}, kh) &= \sum_{\boldsymbol{x} \in \mathbb{Z}^3} f_0(\boldsymbol{x})\phi((kh)^{-1}(\boldsymbol{x} - k\boldsymbol{y})) \\
&= \sum_{\boldsymbol{x} \in \mathbb{Z}^3} f_0(\boldsymbol{x})\phi(h^{-1}(k^{-1}\boldsymbol{x} - \boldsymbol{y})) \\
&= \sum_{k\boldsymbol{x} \in \mathbb{Z}^3} f_0(k\boldsymbol{x})\phi(h^{-1}(\boldsymbol{x} - \boldsymbol{y}))
\end{aligned}
$$

Thus, $f_1(\boldsymbol{y}, h) = f_1(k\boldsymbol{y}, kh)$ only when $f_0(k\boldsymbol{x}) = f_0(\boldsymbol{x})$.

We denote $f_2(\boldsymbol{z}, k)$ as the output of the group convolution layer and $\psi$ as the kernel.

$$f_2(\boldsymbol{z}, k) = \sum_{\boldsymbol{y} \in \mathbb{Z}^3} \sum_{h \in G} f_1(\boldsymbol{y}, h)\psi(k^{-1}(\boldsymbol{y} - \boldsymbol{z}), k^{-1}h)$$

Now we prove that, $f_2(g\boldsymbol{z}, g) = f_2(\boldsymbol{z}, e), \ g \in G$ only when $g$ stabilizes $f_0$, where $e$ is the identity.

$$
\begin{aligned}
f_2(g\boldsymbol{z}, g) &= \sum_{\boldsymbol{y} \in \mathbb{Z}^3} \sum_{h \in G} f_1(\boldsymbol{y}, h)\psi(g^{-1}(\boldsymbol{y} - g\boldsymbol{z}), g^{-1}h) \\
&= \sum_{g\boldsymbol{y} \in \mathbb{Z}^3} \sum_{gh \in G} f_1(g\boldsymbol{y}, gh)\psi(\boldsymbol{y} - \boldsymbol{z}), h)
\end{aligned}
$$

Thus, $f_2(g\boldsymbol{z}, g) = f_2(\boldsymbol{z}, e)$ only when $f_1(g\boldsymbol{y}, gh) = f_1(\boldsymbol{y}, h)$, which means $f_0$ needs to be stablized by $g$ given the previous step of the proof.

Let $Y(\boldsymbol{z})$ and $\hat{Y}(\boldsymbol{z})$, $\boldsymbol{z} \in \mathbb{Z}^3$, be the target and prediction respectively and $L$ is MSE loss. In the last layer, we usually average over the $H$-axis and we can define $\hat{Y}(\boldsymbol{x})$ based on the definition of relaxed group convolution as follows:

$$\hat{Y}(\boldsymbol{z}) = \sum_{k \in H} w(k) f_2(\boldsymbol{z}, k)$$

where $w(k)$ are the relaxed weights. We use MSE loss:

$$L = \sum_{\boldsymbol{z} \in \mathbb{Z}^3} (Y(\boldsymbol{z}) - \hat{Y}(\boldsymbol{z}))^2$$

Finally, we can compute the gradient of the loss $L$ w.r.t a relaxed weight $w(k)$:

$$\frac{\partial L}{\partial w(k)} = \sum_{\mathbf{z} \in \mathbb{Z}^3} \frac{\partial L}{\partial \hat{Y}(\mathbf{z})} \frac{\partial \hat{Y}(\mathbf{z})}{\partial w(k)}$$

$$= -2 \sum_{\mathbf{z} \in \mathbb{Z}^3} |Y(\mathbf{z}) - \sum_{t \in H} f_2(\mathbf{z}, t)| f_2(\mathbf{z}, k)$$

$$= -2 \sum_{k\mathbf{z} \in \mathbb{Z}^3} |Y(k\mathbf{z}) - \sum_{t \in H} f_2(k\mathbf{z}, t)| f_2(k\mathbf{z}, k)$$

This means if $k$ does not stabilize the input $f_0$, then $f_2(\mathbf{kz}, k) \neq f_2(\mathbf{z}, e)$, i.e. $\frac{\partial L}{\partial w(k)} \neq \frac{\partial L}{\partial w(e)}$

If $k$ stabilizes the input $f_0$, then

$$\frac{\partial L}{\partial w(k)} = -2 \sum_{k\mathbf{z} \in \mathbb{Z}^3} (Y(k\mathbf{z}) - \sum_{t \in H} f_2(\mathbf{z}, k^{-1}t)) f_2(\mathbf{z}, e)$$

$$= -2 \sum_{\mathbf{z} \in \mathbb{Z}^3} (Y(k\mathbf{z}) - \sum_{t \in H} f_2(\mathbf{z}, t)) f_2(\mathbf{z}, e)$$

Then we can see $\frac{\partial L}{\partial w(k)} = \frac{\partial L}{\partial w(e)}$ only when $Y(k\mathbf{z}) = Y(\mathbf{z})$.

In conclusion, $\frac{\partial L}{\partial w(k)} = \frac{\partial L}{\partial w(e)}$ only when $k$ stabilizes both the input $f_0$ and the target $Y$. $\qquad \square$

