# OpenReview forum: "Relaxed Octahedral Group Convolution for Learning Symmetry Breaking in 3D Physical Systems"
_NeurIPS.cc/2023/Workshop/AI4Science — NeurIPS2023-AI4Science Poster_

### Official Review · Reviewer_aJVL · 2023-10-16
**Robust but lacking in novelty**

**Rating:** 5
**Confidence:** 2

**Review:**

The paper is well-organized and easy to follow. The introduction provides a clear motivation for the work, and the subsequent sections explain the proposed approach. The paper provides a valuable contribution to the field of deep learning for modeling 3D physical systems. The proposed approach has the potential to significantly improve the accuracy and efficiency of deep learning models in a wide range of applications. However, the paper could benefit from more detailed explanations of some of the technical concepts, particularly for readers who are not familiar with the field. Additionally, the authors should address the issues mentioned in the review comments to improve the clarity and impact of their work.

1. While the implementation of relaxed octahedral group convolution is mathematically robust, the paper lacks a novel idea compared to references in other fields. The relaxed octahedral group convolution (3.1) and separable octahedral group convolution are only the generalized versions of the Eqn. 5 of ref 50 and ref 26 of Eqn. 4, respectively. The main difference between baselines is the relaxed group convolution. However, it is also similar to Relaxed G-Steerable 2D Convolution of ref 50. It would be beneficial for the authors to provide more context and explain how their approach differs from existing techniques.

2. The paper does not show a significant improvement compared to baselines, especially the equivariant model, on the super-resolution of channel flow and isotropic flow. The authors should address this issue and provide a defense for their approach. Moreover, the discussion section is too short, only 6 lines, to defense the lack of a paper.

 3. The paper uses several abbreviations without proper descriptions, which can be confusing for readers. The authors should list all abbreviations used in the paper and explain their meanings. Some examples of abbreviations used in the paper include O and MAE.

4. The paper contains several unofficial expressions, which can be distracting for readers. For example, authors write the reference number as a component of sentences (L25, 26, 27, 30, 33, 158, 160, 166, 171, 173, 175, 178, 180, 188, and maybe more). It is better to directly mention the authors of reference papers for the main reference (e.g., Kondor et al. propose O.). In normal cases, use the complete sentence and just add a reference at the end of the sentence.

The paper presents an interesting approach for modeling 3D physical systems using relaxed octahedral group convolution. However, the authors should address the issues mentioned above to improve the clarity and impact of their work.

---

### Meta-Review · Area_Chair_89WM · 2023-10-26

**Recommendation:** Accept (Poster)
**Confidence:** 3

**Metareview:**

This paper presents a convolution technique for modeling 3D physical systems. It emphasizes that while deep equivariant models can be sample-efficient, assuming perfect symmetry can be limiting. The introduced method aims to balance maintaining equivariance with identifying symmetry-breaking factors in the data. Empirical evidence suggests the technique's capability in understanding symmetry-breaking elements in phase transitions and performing well in fluid super-resolution tasks.

The reviewer appreciates the clarity and structure of the paper and acknowledges its potential contribution to 3D physical system modeling. They recognize the potential for improved accuracy and efficiency in multiple applications through this approach.

However, concerns include:

- The need for more detailed explanations for those unfamiliar with the subject.
- The perceived lack of novelty when compared to other referenced work, suggesting the new approach may just be a generalization of existing methods.
- Inadequate differentiation from baseline models in specific task results.
- Abbreviations are used without explanations, leading to potential confusion.

While there are concerns to address, the paper's motivation and its contribution to the field are good to be accepted. It introduces an approach that can be impactful for 3D physical system modeling.